# Vascular and metabolic risk factor differences prior to dementia diagnosis: a multidatabase case–control study using European electronic health records

Gayan Perera [1], P R Rijnbeek,[2] Myriam Alexander,[3] David Ansell,[4] Paul Avillach,[5,6] Talita Duarte-Salles [7], Mark Forrest Gordon,[8] Francesco Lapi,[9] Miguel Angel Mayer,[10] Alessandro Pasqua,[9] Lars Pedersen,[11] Johan van Der Lei,[2] Pieter Jelle Visser,[12,13,14] Robert Stewart[1,15]

► Prepublication history and additional materials for this paper is available online. To view these files, please visit the journal online (http://dx.doi.org/10.1136/bmjopen-2020-038753).

For numbered affiliations see end of article.

**Correspondence to**
Dr Robert Stewart;
robert.stewart@kcl.ac.uk

## ABSTRACT

**Objective** The objective of the study is to compare body mass index (BMI), systolic/diastolic blood pressure (SBP/DBP) and serum total cholesterol levels between dementia cases and controls at multiple time intervals prior to dementia onset, and to test time interval as a modifying factor for these associations.

**Design** Case–control study.

**Setting** Six European electronic health records databases.

**Participants** 291 780 cases at the date of first-recorded dementia diagnosis, compared with 29 170 549 controls randomly selected from the same databases, age matched and sex matched at this index date.

**Exposure** The following measures were extracted whenever recorded within each dataset: BMI (kg/m$^2$), SBP and DBP (mm Hg) and serum total cholesterol (mmol/L). Levels for each of these variables were defined within six 2-year time intervals over the 12 years prior to the index date.

**Main outcomes** Case–control differences in exposures of interest were modelled for each time period and adjusted for demographic and clinical factors (ischaemic/unspecified stroke, type 2 diabetes mellitus, acute myocardial infarction, hypertension diagnosis, antihypertensive medication, cholesterol-lowering medication). Coefficients and interactions with time period were meta-analysed across the six databases.

**Results** Mean BMI (coefficient −1.16 kg/m$^2$; 95% CI −1.38 to 0.93) and SBP (−2.83 mm Hg; 95% CI −4.49 to −1.16) were lower in cases at diagnosis, and case–control differences were greater in more recent time periods, as indicated by significant case-x-time interaction and case-x-time-squared interaction terms. Time variations in coefficients for cholesterol levels were less consistent between databases and those for DBP were largely not significant.

**Conclusion** Routine clinical data show emerging divergence in levels of BMI and SBP prior to the diagnosis of dementia but less evidence for DBP or total cholesterol levels. These divergences should receive at least some consideration in routine dementia risk

## Strengths and limitations of this study

► We have investigated associations of blood pressure, body mass index and cholesterol levels with dementia and the time dependency of such observations, using what we believe to be currently the largest confederation of health records data.

► There are advantages in the generalisability of routine data and in patient populations who are less subject to participation selection than research cohorts.

► The use of multiple data resources allowed the consistency of associations to be evaluated, as it is particularly important for clinical 'big data' where large sample sizes result in even very minor associations being identified as statistically significant, and where traditional heterogeneity statistics from meta-analyses are similarly uninformative.

► The limitations of the study are those that typify routine administrative rather than research data, in that measurements were limited to those recorded in routine practice which could be harmonised across the six databases, and the stage of dementia at diagnosis was not known.

► While the large samples will have reduced the impact of missing data on statistical power, potential bias arising from non-random missingness cannot be excluded, although we feel it is unlikely to account for the findings of interest.

screening, although underlying mechanisms still require further investigation.

## INTRODUCTION

Vascular risk factors are widely recognised as important in the aetiology of dementia; however, observed associations depend on the timing of measurement. For example, higher blood pressure (BP) (particularly



systolic BP; SBP) in mid-life is a predictor of dementia 10–20 years later,[1–3] but people with dementia tend to have lower BP at the time of the clinical onset than age-matched controls,[1] accounted for by an exaggerated decline in SBP prior to the onset of dementia.[4–6] Similar associations have been observed for body mass index (BMI), with mid-life obesity associated with increased risk of dementia,[7 8] but dementia clinical onset associated with lower contemporaneous BMI[9] and preceded by accelerated weight loss.[10] Finally, higher mid-life cholesterol has been found to predict dementia in some studies,[11] and a decline in cholesterol levels prior to dementia onset has been observed in others.[12 13]

These findings have arisen almost exclusively from specifically recruited research cohorts followed over long periods—advantageous in many respects but potentially subject to selection, and therefore limited generalisability. The parameters of interest (BP, BMI, cholesterol levels) are routinely measured in clinical practice, but there has been no investigation of their associations with dementia and the time-dependency of such observations. We sought to pursue this in what we believe to be currently the largest confederation of health records data—that enabled by the European Medical Information Framework Innovative Medicines Initiative (EMIF-IMI) within which an Alzheimer's Disease (EMIF-AD) consortium was developed. The objective of the study reported here was to compare levels of the exposures of interest between dementia cases and controls at multiple time intervals prior to dementia onset, and to test time interval as a modifying factor for these associations. The specific hypothesis was that BMI, SBP and diastolic BP (DBP) and serum total cholesterol would be lower in people with newly diagnosed dementia than in controls, and that these differences would have widened closer to the diagnosis date. In this study, we used data from several European national or regional databases—both primary care general practitioner (GP) records and hospital based databases—containing a total number of over 22.5 million patients who used mainly public/national health services.

## METHODS
### Data resources
EMIF-IMI worked with several electronic health record (EHR) resources to render data available for analysis with appropriately robust data security and governance. EMIF datasets participating in a recent investigation of incidence and prevalence of diagnosed dementia were estimated to contain information on 182 776 people with a dementia diagnosis, with around 1 000 000 person-years of data prior to the diagnosis and 400 000 person-years after the diagnosis.[14] These resources included three primary care databases from the Netherlands, Italy and UK (Integrated Primary Care Information (IPCI), Health Search Database (HSD) and The Health Improvement Network (THIN) respectively) and two hospital catchment area databases in Jutland and Barcelona (Aarhus

University Hospital Database (AUH) and Information System of Parc de Salut Mar (IMIM-UPF), respectively). In this study, the 'Information System for Research in Primary Care (SIDIAP), containing Catalan primary care records) was added as a new database. The final assembled data resource included a total 291 780 patients with a dementia diagnosis across the six databases.

The THIN database contains data from participating UK primary care services with records from 1990 on around 9 million patients in total and 4 million receiving active care at a given time. Demographic profiles have been found to be similar to national estimates although THIN contains fewer people aged under 25 years; condition prevalence has also been found to be comparable.[15] The IPCI database is drawn from primary care records in Netherlands since 1990 on around 2.6 million patients in total, and 1.4 million receiving active care; its data are drawn from what has been found to be a representative sample (n=750) of GPs in the Netherlands.[16] AUH is a regional record linkage system in Aarhus, Denmark containing both inpatient and outpatient records since 1989 with 1.8 million active patients, drawing data from different public bodies and including all ages.[17] IMIM-UPF contains hospital data from the Barcelona region with 1.6 million active patients and including all hospital admissions. HSD contains primary care records in Italy with nearly 2.3 million patient records from more than 800 participating primary care services throughout Italy: a longitudinal observational database that has been found to be representative of the general Italian population, established in 1998 by the Italian College of General Practitioners.[18] SIDIAP contains primary care records from Catalonia in Spain with nearly 6 million patient records from family paediatricians and GP medical records and data from all ages.[17]

The EMIF project, funded through the IMI programme (Innovative Medicines Initiative Joint Undertaking under grant agreement no. 115372), had designed and implemented a federated platform to connect health data from a variety of sources across Europe, to facilitate large scale clinical and life sciences research. It enabled approved users to analyse securely multiple, diverse, data via a single portal, thereby mediating research opportunities across a large quantity of research data. EMIF developed a code of practice (ECoP) to ensure the privacy protection of data subjects, protect the interests of data sharing parties, comply with legislation and various organisational policies on data protection, uphold best practices in the protection of personal privacy and information governance, and eventually promote these best practices more widely. EMIF convened an Ethics Advisory Board, to provide feedback on its approach, platform, and the EcoP[19] and all use of the data resources described was carried out with full approval of the data owners and thus complied with pre-existing governance specifications for secondary analysis.

This study created an age-matched and sex-matched case–control study, comparing people with and without

dementia on prior exposures of interest over the preceding 12 years, adjusting for relevant and available covariates, and investigating time interval as a modifying factor.

## Population

Cases were defined on the basis of a diagnosis of dementia and included all assigned dementia diagnoses of any type (inclusion criterion) and the index date was set as that on which the first dementia diagnosis was recorded. The definition of dementia was extensively developed and harmonised across the EMIF-AD consortium, taking into account the different coding systems used, as previously described in detail including supplementary data on code lists applied.[14] For each case, at the index date of first dementia diagnosis, 100 controls were drawn randomly from the same EHR, matched to the case on gender and age (±2 years) at that index date, having excluded any dementia at or prior to that date. When there were not enough controls, one control was allowed to be used for several cases. Both cases and controls were restricted to people aged 50 years or over at the index date.

## Exposures

The following measures were extracted whenever recorded within each dataset: BMI ($kg/m^2$), SBP (mm Hg), DBP (mm Hg) and serum total cholesterol (mmol/L). BMI was extracted where recorded, or was calculated from recorded body weight and height data. Levels for each of these variables were defined within six 2-year time intervals over the 12 years prior to the index date, taking mean values if multiple measures were present for an individual within each interval. Percentages of missing data points were compared between cases and controls, and by age, sex and comorbidities.

## Covariates

Data on age at index date and gender were extracted. In addition, the following binary variables were generated based on any recorded diagnosis/treatment prior to the index date: ischaemic/unspecified stroke, type 2 diabetes mellitus (diagnosis or antidiabetic medication), acute myocardial infarction, hypertension diagnosis, use of any antihypertensive medication and use of any cholesterol-lowering medication. Hypertension diagnosis was considered as a relevant covariate as patients with sustained hypertension in midlife have an increased risk for dementia regardless of their BP during late life[20] and because the interest was on changes in the BP relationship over time, which was not considered to be coterminous with a hypertension diagnosis. Medication information was extracted from databases in which it was available from recorded prescriptions or dispensations; disorders were defined according to pre-existing harmonised data extraction criteria used across the EMIF-IMI consortia.[21]

## Statistical analysis

The total pool of cases and controls were first described in terms of demographic and clinical covariates, followed by a descriptive analysis of data availability on the exposures of interest.

General linear model (GLM) analyses were used for each database and for each of the four parameters of interest, to model case–control differences in mean exposure levels separately for each comparison time period and to adjust for covariates. Importantly, each time period comparison was treated as a separate analysis for case–control sets with data present in that time period and covariate adjustments for each comparison, rather than in terms of repeated observations within individuals. The exposures here were modelled as dependent variables, with main independent variables being case–control status, time category, interaction between case–control status and time category, age at index date, sex, the four vascular disorders described above and stratum variable. Based on plots of unadjusted mean differences by time period, linear time terms were used for DBP as a dependent variable while quadratic time terms were added to models of BMI, SBP and total cholesterol.

Having finalised models for each of the four parameters (BMI, SBP, DBP, total cholesterol) in each EHR dataset, the GLM coefficients for each dataset were pooled in a meta-analysis for each parameter. Random effects were assumed when coefficients were pooled. In secondary analyses, adjusted meta-analysed coefficients for SBP, DBP and total cholesterol were stratified as appropriate according to previously received antihypertensive or lipid lowering medication in cases and controls. The significance level for all tests was defined as a $p \leq 0.05$. Analyses were carried out using IBM SPSS Statistics V.20.0.

## Patient and public involvement

The was no specific patient or public involvement in the design or interpretation of this particular study.

## RESULTS

Overall, there were 291 780 dementia cases (ranging from 3433 cases in IMIM-UPF to 139 083 cases in THIN) and 29 170 549 controls made available for analysis from all six EHR databases. As described in table 1, age and sex were effectively matched in cases and controls. Higher proportions of cases had a history of diabetes, acute myocardial infarction and stroke. Slightly higher proportions of cases had been prescribed antihypertensive, as well as lipid lowering, medication than controls; however, prevalence of recorded hypertension did not differ consistently between cases than controls across the databases. Descriptive data on mean vascular risk factor levels by case/control status, time period and database are provided in online supplemental table 1, missing data percentages by year and case status in online supplemental table 2, and missing data by covariates in online supplemental table 3. Missing data were more prominent for less recent time periods; for time periods over 2 years before the index date, missing data were more common in cases than controls, but more common in controls than cases for the

**Table 1** Characteristics of component databases: demographic factors and comorbidities

| | AUH | | HSD | | IPCI | | SIDIAP | | THIN | | IMIM-UPF | |
|---|---|---|---|---|---|---|---|---|---|---|---|---|
| | Case | Control | Case | Control | Case | Control | Case | Control | Case | Control | Case | Control |
| No | 4756 | 475319 | 21465 | 2145779 | 14039 | 1401114 | 109004 | 10900377 | 139083 | 13907094 | 3433 | 340866 |
| Mean (SD) age | 78.8 (9.7) | 78.7 (9.7) | 80.6 (8.3) | 80.4 (8.2) | 81.1 (8.7) | 80.9 (8.6) | 81.4 (8.1) | 81.3 (8.1) | 80.9 (8.9) | 80.6 (8.9) | 80.0 (9.6) | 79.8 (9.5) |
| Males (%) | 43.0 | 43.0 | 32.8 | 32.8 | 36.7 | 36.7 | 33.8 | 33.8 | 34.8 | 34.8 | 42.1 | 42.0 |
| Medication use (%) | | | | | | | | | | | | |
| Antihypertensive | 79.1 | 78.2 | 76.1 | 72.9 | 75.0 | 75.4 | 77.2 | 73.8 | 56.6 | 63.8 | n/a | n/a |
| Lipid lowering | 47.8 | 45.0 | 26.9 | 26.1 | 42.3 | 39.4 | 46.6 | 42.2 | n/a | n/a | n/a | n/a |
| Comorbidities (%) | | | | | | | | | | | | |
| Diabetes | 13.0 | 9.5 | 23.1 | 19.8 | 21.2 | 18.9 | 26.6 | 18.9 | 13.0 | 10.7 | 29.7 | 14.0 |
| Hypertension | 40.3 | 33.6 | 63.9 | 61.0 | 48.6 | 52.2 | 64.5 | 64.2 | 49.6 | 52.5 | 61.0 | 37.9 |
| Acute MI | 6.7 | 7.3 | 2.4 | 2.1 | 7.5 | 6.4 | 3.4 | 2.6 | 8.2 | 5.9 | 4.2 | 2.7 |
| Stroke | 13.8 | 7.8 | 8.3 | 6.5 | 12.4 | 7.7 | 12.3 | 6.9 | 10.6 | 5.7 | 12.8 | 3.9 |

AUH, Aarhus University Hospital; HSD, Health Search Database; IMIM-UPF, Information System of Parc de Salut Mar; IPCI, Integrated Primary Care Information; MI, myocardial infarction; n/a, data not available; SIDIAP, Information System for Research in Primary Care; THIN, The Health Improvement Network.

0–2 years time period; missing data were least common for SBP/DBP and most common for BMI. Mean numbers of missing data points were not found to vary consistently by age or sex, although missing data were consistently below mean levels in those with hypertension and above mean levels in those with stroke. Online supplemental file 1 summarises proportions of patients with missing data for each vascular risk factor in each database. In summary, AUH had limited BMI data, with no values prior to 6 years before the index date and no data for SBP and DBP. For HSD, 94% of BMI data were missing in the 10–12 years window before the index date, improving to 78% by the 0–2 years window and similar trends were observed for BMI data in the IPCI database, with 93% missing at 10–12 years and 67% missing data at 0–2 years before the index date. In the SIDIAP database there were no BMI, BP or total cholesterol data at 10–12 years before the index date. Generally, compared with other databases, there were fewer missing data in the THIN database for all risk factors. IMM-UPF did not have BP data, and had no BMI data at 10–12 years before the index date.

Figure 1 displays case–control differences in vascular risk factor exposures of interest by time and database. For BMI, the largest decline in meta-analysed case–control differences was observed between 2–4 years and 0–2 years before the index diagnosis of dementia. For SBP, the meta-analysed case–control difference increased from 8–10 years to 4–6 years and then declined steeply from 2–4 years to 0–2 years. For DBP, the largest meta-analysed case–control differences were observed in the 2–4 years and 0–2 years periods before the diagnosis of dementia. For total serum cholesterol levels, the meta-analysed case–control difference increased from 10–12 years to 6–8 years before the diagnosis of dementia and then declined up till the diagnosis of dementia with a slightly large decline just before diagnosis of dementia. For all exposures apart from DBP, cases had lower values compared with controls (negative case coefficients), all four exposures decreased in later time periods (negative time coefficients) and this was more marked for cases compared with controls (negative case-x-time interaction terms) (table 2). Quadratic terms were also significant and negative in value for BMI and SBP, reflecting the curvilinear relationships displayed in figure 1, while those for total cholesterol comprised a positive time-squared term and a negative case-x-time-squared interaction, again reflecting the curvilinear visualised relationship. Following adjustment for covariates, case-x-time and case-x-time-squared interaction terms were reduced modestly for BMI and SBP but retained statistical significance. No significant case or case-x-time terms were observed for DBP. For total cholesterol a negative-value case-x-time interaction term persisted, but there was no longer a significant case-x-time-squared interaction.

Following additional stratification of these adjusted models by medication use (table 3), SBP coefficients were not markedly different (overlapping confidence intervals), although those for case status and the case-x-time

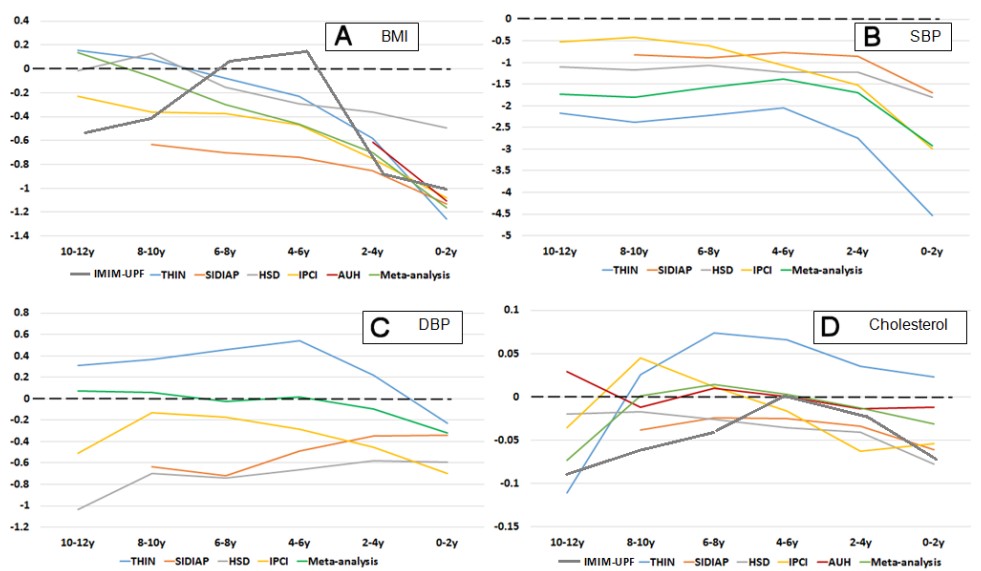

**Figure 1** Mean differences between cases and controls in vascular risk factor exposures of interest by database and time prior to index date. AUH, Aarhus University Hospital Database; BMI, body mass index; DBP, diastolic blood pressure; HSD, Health Search Database; IPCI, Integrated Primary Care Information; IMIM-UPF, Information System of Parc de Salut Mar; SBP, systolic blood pressure; SIDIAP, Information System for Research in Primary Care; THIN, The Health Improvement Network.

interaction were stronger in the subgroups who were not receiving antihypertensive medication; the (negative value) case coefficient for DBP was also stronger in that subgroup. Likewise, case and case-x-time coefficients for total cholesterol levels were stronger in those not receiving lipid lowering medication, although CIs overlapped.

## DISCUSSION

In this large study spanning 12 years, we compared people with/without a dementia diagnosis and investigated previously recorded levels of BMI, SBP, DBP and total serum cholesterol. In summary, BMI, SBP and total cholesterol levels were lower overall in cases with dementia than

**Table 2** Meta-analysed unadjusted GLM coefficients (with 95% CIs) across component databases for the relationships between case/control status, time period and the four outcomes of interest

|  | BMI | SBP* | DBP*† | Cholesterol |
|---|---|---|---|---|
| **Unadjusted** |  |  |  |  |
| Intercept | 31.6 (31.6 to 31.6) | 114.6 (114.5 to 114.6) | 84.4 (84.4 to 84.4) | 5.4 (5.3 to 5.5) |
| Case | −1.32 (−1.41 to −1.21) | −3.25 (−3.34 to −3.16) | 0.04 (−0.03 to 0.11) | −0.12 (−0.20 to −0.04) |
| Time | −0.11 (−0.17 to −0.06) | −1.06 (−1.17 to −0.98) | −0.54 (−0.61 to −0.45) | −0.03 (−0.05 to −0.01) |
| Case-x-time | −0.27 (−0.32 to −0.19) | −0.63 (−0.72 to −0.57) | −0.03 (−0.06 to −0.01) | −0.03 (−0.04 to −0.02) |
| Time squared | −0.06 (−0.10 to −0.02) | −0.13 (−0.19 to −0.07) |  | 0.01 (0.00 to 0.02) |
| Case-x-time squared | −0.02 (−0.03 to −0.01) | −0.14 (−0.21 to −0.10) |  | −0.01 (−0.01 to −0.00) |
| **Adjusted‡** |  |  |  |  |
| Intercept | 31.5 (30.9 to 32.2) | 117.6 (108.8 to 126.5) | 87.0 (81.7 to 92.3) | 5.18 (4.89 to 5.50) |
| Case | −1.16 (−1.38 to −0.93) | −2.83 (−4.49 to −1.16) | −0.19 (−0.46 to 0.08) | −0.10 (−0.17 to -0.04) |
| Time | −0.13 (−0.16 to −0.10) | −0.83 (−1.16 to −0.50) | −0.50 (−0.57 to −0.42) | −0.05 (−0.07 to −0.02) |
| Case-x-time | −0.18 (−0.28 to −0.08) | −0.53 (−0.80 to −0.26) | 0.00 (−0.06 to 0.06) | −0.05 (−0.09 to −0.01) |
| Time squared | −0.01 (−0.08 to 0.06) | −0.01 (−0.02 to 0.02) |  | 0.00 (−0.02 to 0.02) |
| Case-x-time squared | −0.01 (−0.02 to −0.00) | −0.04 (−0.05 to −0.02) |  | −0.01 (−0.01 to 0.01) |

*SBP and DBP not available in AUH or IMIM-UPF.
†Based on linear equation.
‡GLM coefficients adjusted for age, sex, diabetes, hypertension, acute myocardial infarction and stroke.
AUH, Aarhus University Hospital; BMI, body mass index; GLM, general linear model; IMIM-UPF, Information System of Parc de Salut Mar; SBP/DBP, systolic/diastolic blood pressure.

**Table 3** Meta-analysed adjusted GLM coefficients stratified by antihypertensive treatment and lipid lowering medication

| SBP | With antihypertensive treatment | No antihypertensive treatment |
|---|---|---|
| Intercept | 121.6 (115.5 to 127.8) | 109.4 (101.4 to 117.5) |
| Case | −2.44 (−3.56 to −1.33) | −4.02 (−8.67 to 0.63) |
| Time | −1.01 (−1.40 to −0.62) | −0.14 (−0.55 to 0.28) |
| Case-x-time | −0.49 (−0.60 to −0.32) | −0.72 (−1.54 to 0.10) |
| Time squared | −0.03 (−0.04 to −0.01) | −0.04 (−0.05 to −0.02) |
| Case-x-time squared | −0.04 (−0.05 to −0.03) | −0.04 (−0.08 to 0.01) |
| **DBP** | **With antihypertensive treatment** | **No antihypertensive treatment** |
| Intercept | 89.3 (85.8 to 92.9) | 82.3 (76.1 to 88.5) |
| Case | −0.05 (−0.43 to 0.33) | −1.07 (−1.57 to −0.56) |
| Time | −0.55 (−0.64 to −0.50) | −0.15 (−0.22 to −0.08) |
| Case-x-time | 0.02 (−0.02 to 0.07) | −0.11 (−0.29 to 0.07) |
| **Cholesterol** | **With lipid-lowering drugs** | **No lipid-lowering drugs** |
| Intercept | 5.23 (5.20 to 5.25) | 5.44 (5.18 to 5.71) |
| Case | 0.02 (−0.01 to 0.05) | −0.08 (−0.12 to −0.03) |
| Time | −0.07 (−0.10 to −0.04) | −0.02 (−0.06 to 0.02) |
| Case-x-time | −0.01 (−0.01 to 0.01) | −0.03 (−0.04 to −0.02) |
| Time squared | −0.03 (−0.04 to −0.01) | −0.04 (−0.05 to −0.02) |
| Case-x-time squared | −0.01 (−0.02 to 0.00) | −0.01 (−0.01 to −0.00) |

GLM, general linear model; SBP/DBP, systolic/diastolic blood pressure.

controls and the difference was most marked closest to the point of diagnosis. This time dependency of the associations was not accounted for by age, sex or recorded vascular disorders, and case–control differences in SBP and total cholesterol did not appear to be modified by antihypertensive or lipid-lowering treatment respectively. There was little evidence for associations with DBP after adjustment, and between-database heterogeneity appeared high for associations with total cholesterol, unlike the relative consistency observed with BMI and SBP. As described, while dementia is generally found to be associated with higher levels of vascular risk factors measured 10–20 years before its clinical onset in prospective epidemiological studies, associations are often found to be reversed when investigated closer to time of this clinical onset.[22] The extent to which these associations can be observed in more healthcare data remains to be established, despite opportunities for assessing this over large, 'real-world' populations.

Considering BMI, prospective studies with decades follow-up[23] have suggested that midlife obesity is a risk factor for dementia,[24] but this association is attenuated or reversed in older age,[25] and exaggerated weight loss precedes the clinical onset of dementia,[26] described for periods of 2–4 years,[27] 5 years[28] or up to 10 years[29] before dementia onset. Our findings of lower SBP in cases at the time of diagnosis, and a preceding widening difference over time, are also consistent with epidemiological research, as is the lack of association with DBP: high

mid-life SBP is a risk factor for dementia incidence 10–20 years later,[2 3 5] while SBP at dementia clinical onset tends to be lower than controls,[5 30] and exaggerated decline in SBP prior to dementia has been described in Swedish[4 31] and Japanese American[6] cohorts, as well as during the course of dementia.[32] Although case and case-x-time coefficients were statistically significant for total cholesterol levels, similar to those for BMI and SBP, visual inspection (figure 1) indicated considerably more inconsistency between databases and we feel that conclusions are less strong in this respect. The relationship between cholesterol level and dementia has been less clear in epidemiological research,[33] although tends to support associations with high total cholesterol mid-life but not in late-life. Declining cholesterol levels precede dementia,[12 13] although without the later acceleration described for BMI and SBP, and not observed in all studies.[34]

Assuming that the widening gaps in exposure levels between cases and controls in our data reflect dementia-associated decline in these factors, underlying causal pathways may vary. Weight loss might be secondary to early neurodegeneration, reflecting a combination of predementia apathy, loss of initiative and reduced olfactory function.[29] Cholesterol decline in one study was observed to occur over 10 years before dementia clinical onset, followed by a parallel trajectory,[13] so could conceivably be a marker for another risk factor, such as an episode of infection or inflammation resulting in both the decline in cholesterol and increased later risk of dementia. While declining BP might be secondary to early neurodegeneration, it might also represent a risk factor in itself, through reduced cerebral perfusion and watershed ischaemia/infarction, and/or might be a consequence of previous untreated hypertension. The role of antihypertensive treatment in accounting for or modifying predementia blood pressure decline has been controversial. In one cohort, the exaggerated dementia-associated decline in SBP was found to be restricted to people who had not received antihypertensive agents,[6] and associations between dementia and lower contemporaneous BP were found to be weak or absent in populations from low-income and middle-income countries where hypertension is uncommon.[35] However, other cohorts have found stronger contemporaneous associations between dementia and lower BP in people receiving antihypertensives[36] or no modification of later SBP decline.[31] In our databases, the coefficients of interest (ie, for case and case-x-time) were stronger in the subgroup who had not received antihypertensive treatment, but differences were not substantial and confidence intervals overlapped. The same was true for associations with cholesterol levels, which were also marginally stronger in the subgroup who had not received lipid lowering treatment, although no attempt was made to factor in treatment differences in lipid-lowering effectiveness.

In previous research, there has been some suggestion of an age-dependent effect of BP on AD, although the evidence is relatively weak: there was a suggestion that

mid-life diastolic, but not midlife systolic, hypertension in midlife may increase risk of incident AD and a suggestion that elevated late-life BP may actually be beneficial.[37] However, the same systematic review[37] and meta-analysis of prospective epidemiological research, did not provide clear evidence for a relationship between BP and incidence of AD. On the other hand,[38] higher SBP was found among patients with vascular dementia (OR per 10 mm Hg: 1.33). Several well-established longitudinal studies have reported on the relationships between the trajectory of change over time in levels of BP, cholesterol and BMI and incident all-cause dementia, with some additionally reporting on AD, vascular dementia and MCI.[39] Authors found that the results for AD, vascular dementia and MCI were similar but with fewer data points, with those who went on to develop all-cause dementia showing a greater increase, followed by a sharper decrease in BP and/or BMI before a positive diagnosis.

In a collaborative study of over 1.3 million adults from Europe, the USA and Asia, higher BMI was associated with increased dementia risk when weight was measured >20 years before dementia diagnosis,[40] but this association was reversed when BMI was assessed <10 years before dementia diagnosis and this finding is consistent with our study as well as a systematic review.[41] Furthermore, comparisons of risk factors for dementia, AD and VaD, in developed and developing world regions were carried out in a review by Kalaria in 2010 which concluded that there was a high risk effect of factors such as vascular diseases and smoking for incidence of dementia in developed regions (North America, Europe, Japan) as well as in Asia (China, Guam, India, South Korea, Taiwan). Dementia was significantly associated with reported stroke and diabetes in logistic regression models adjusted for socio-demographic status and other vascular risk factors (OR 4.40 (95% CI 2.70 to 7.19) and OR 1.56 (95% CI 1.20 to 2.03), respectively) in a national survey of older people in Trinidad.[42]

Information on education level was not available in these routine healthcare data resources. On the one hand, low educational achievement has been shown to be a robust risk factor for dementia[43]; on the other hand, intellectually stimulating, socially engaging, or physical activities might lower the risk of dementia.[44] The situation is not different in low/middle-income countries, where surveys have consistently identified low education as a risk factor for dementia.[45] However, in some communities, low literacy is often linked to poverty or lower socioeconomic status, which is also associated with poorer health, lower access to healthcare and increased risk of dementia.[45] While it may well be the case that lower education confers both a higher risk of dementia and a higher risk of adverse vascular risk factors, it is less clear whether there should be any association between education and the changes over time in relationships between vascular risk factors and dementia, described in this study.

Our study has several strengths. We believe that the data resource used for this analysis is the largest deployed to date in dementia research by a considerable margin, illustrating the potential value of routine healthcare data and increasing accessibility of EHRs for research use within robust governance frameworks such as that developed by the EMIF-IMI consortium. In addition, there are advantages in the generalisability of routine data and in patient populations who are less subject to participation selection than research cohorts. Finally, the use of multiple data resources allowed the consistency of associations to be evaluated. This is particularly important for clinical 'big data' where large sample sizes result in even very minor associations being identified as statistically significant, and where traditional heterogeneity statistics from meta-analyses are similarly uninformative; this is exemplified in the visualisation of associations with DBP and total cholesterol (figure 1) where reported analyses from single databases might have given rise to conclusions which are not consistent across others.

The limitations of the study are those that typify routine administrative rather than research data. Measurements were limited to those recorded in routine practice which could be harmonised across the six databases, bearing in mind the different clinical coding systems used. For this reason, and as previously described[14] dementia was defined as a composite outcome with no attempt to identify subtypes which are less consistently ascertained in routine clinical practice. Covariates were limited to demographic factors, common disorders and specific medication groups of interest, and did not include measures of socioeconomic status or lifestyle factors such as smoking status, diet, or physical activity. Missing data are also an important consideration. While the large samples will have reduced the impact of missing data on statistical power, potential bias arising from non-random missingness cannot be excluded, although we feel it is unlikely to account for the findings of interest. As displayed in online supplemental table 2, the strongest influence on data loss was time, with more missing data in earlier time periods, as would be expected in clinical samples (where routine exposure checks will be more common in older age groups). Therefore, case–control differences at earliest time points should be viewed with caution. Furthermore, two databases, AUH and IMIM-UPF, did not contribute BP data and IMIM-UPF and HSD had high proportion of missing data for BMI throughout the study period. However, it is hard to envisage how differential missing data could account for lower exposure levels in future cases, or for the widening of case–control differences over time (since this was not observed for all exposures of interest). Furthermore, influences of age, sex and comorbidity on missingness were relatively minor, apart from understandable higher recording in monitored conditions such as hypertension. Finally, although missing data were more common in cases than controls for most time periods, these differences were not substantial. Consideration was given to missing data when analyses

were planned; however, the limited level of covariate ascertainment and potential non-random missingness were felt to preclude imputation approaches.

An important consideration with dementia diagnoses in routine healthcare data from non-specialist services is that there is minimal or no information on the stage/severity of dementia at diagnosis, because of the absence of structured data on cognitive function. While some people will have received a dementia diagnosis early in the course of the condition, others may receive this after several years of significant symptoms. For example, the length of time between dementia symptoms being noticed and diagnosis of dementia has been estimated to vary in Europe from 1.61 years in Italy to 2.49 years in Scotland and 2.57 years in Netherlands.[46] Therefore, comparisons are limited with epidemiological cohorts using dementia screening protocols. At least some of the widening difference in exposure levels between cases and controls may, therefore, have occurred during times when dementia was evident but undiagnosed. Finally, our analyses did not analyse exposure trajectories within individuals, which is the standard approach in epidemiological cohorts. Instead we investigated case–control differences in previously recorded exposures at different times, hypothesising that the dementia-associated trajectories previously reported for these factors would manifest in variation of case–control differences across time periods, and hence case-x-time interaction in the statistical models.

## CONCLUSION

In summary, considering our observations at group level, there are clear potential implications for clinical services which require further evaluation. For example, there might be good reason for individual trajectories to be more effectively monitored (eg, BMI or SBP change), possibly enhanced by EHR systems that could highlight changes over time and the need for more frequent assessment or supplementary cognitive assessment for earlier risk-identification. In addition, regardless of causation, the potential consequences of low BMI/SBP as comorbidities in dementia have yet to be fully clarified (eg, as risk factors for falls or functional decline) which again might have implications for future routine monitoring. In examining the patterns and trajectories of the established risk factors for cognitive decline and dementia, this study adopted a relatively broad approach to an emerging area of enquiry. Our study focused on three risk factors of interest that were envisaged to be most available (BP, cholesterol and BMI); however, future work could feasibly include greater numbers of risk factors and the interaction between different individual changes in status (eg, the combination of changes in BP and BMI), which could eventually lead to more personalised risk assessments and targeted interventions early in the asymptomatic, prodromal phase of cognitive decline and dementia.

**Author affiliations**
[1]Psychological Medicine, King's College London (Institute of Psychiatry, Psychology and Neuroscience), London, UK
[2]Department of Medical Informatics, Erasmus Medical Center, Rotterdam, The Netherlands
[3]Quantitative Sciences, GlaxoSmithKline Plc, Brentford, UK
[4]Institute of Applied Health Research, University of Birmingham, Birmingham, UK
[5]Department of Biomedical Informatics, Harvard Medical School, Boston, Massachusetts, USA
[6]Aarhus University, Aarhus, Denmark
[7]IDIAP Jordi Gol, Barcelona, Spain
[8]Specialty Clinical Development, Neurology and Psychiatry, Teva Pharmaceuticals USA Inc, North Wales, Pennsylvania, USA
[9]Health Search, Italian College of General Practitioners and Primary Care, Florence, Italy
[10]Hospital del Mar Institute for Medical Research, Barcelona, Spain
[11]Department of Clinical Epidemiology, Aarhus University Hospital, Aarhus, Denmark
[12]Department of Psychiatry and Neuropsychology, Maastricht University, Maastricht, The Netherlands
[13]Department of Neurology, Vrije Universiteit, Amsterdam, The Netherlands
[14]Department of Neurobiology, Karolinska Institutet, Stockholm, Sweden
[15]South London and Maudsley NHS Foundation Trust, London, UK

**Acknowledgements** MAA received funding from Roche and GSK as a contracted employee; GP received funding from Roche as contracted employee during this project; MFG has prior employment at Boehringer Ingelheim Pharmaceuticals, and is a current employee of Teva Pharmaceuticals; RS has received research funding from Roche, Janssen, GSK and Takeda.

**Contributors** The analysis was conceived and designed by RS, PRR, PJV, MA, MFG, FL and JvDL. Data extractions and advice on analysis design were provided by the data controllers (DA, TD-S, MAM, AP, LP, JvDL) in liaison with PRR and PA. Analyses were carried out by LP and GP. The manuscript was written by GP and RS. Further comments and significant input were obtained from all coauthors who also oversaw the planning and development of the project.

**Funding** The research leading to these results has received support from the Innovative Medicines Initiative (IMI) Joint Undertaking under EMIF grant agreement n° 115372, resources of which are composed of financial contribution from the European Union's Seventh Framework Programme (FP7/2007-2013) and EFPIA companies' in kind contribution. RS and GP are part-funded by the National Institute for Health Research (NIHR) Biomedical Research Centre and Dementia Biomedical Research Unit at South London and Maudsley NHS Foundation Trust and King's College London.

**Competing interests** None declared.

**Patient consent for publication** Not required.

**Provenance and peer review** Not commissioned; externally peer reviewed.

**Data availability statement** No additional data are available. The datasets analysed during the current study are not publicly available due to confidentially of patients but are available via application to individual data custodians under the EMIF platform (http://www.emif.eu/). The data provided by each database owners were already anonymised.

**ORCID iDs**
Gayan Perera http://orcid.org/0000-0002-3414-303X
Talita Duarte-Salles http://orcid.org/0000-0002-8274-0357

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
