## [Reviewer comments · BMJ Open]

ARTICLE DETAILS

TITLE (PROVISIONAL)	Vascular and metabolic risk factor differences prior to dementia diagnosis: a multi-database case control study using European electronic health records
AUTHORS	Perera, Gayan; Rijnbeek, P R; Alexander, Myriam; Ansell, David; Avillach, Paul; Duarte-Salles, Talita; Gordon, Mark Forrest; Lapi, Francesco; Mayer, Miguel Angel; Pasqua, Alessandro; Pedersen, Lars; van Der Lei, Johan; Visser, Pieter Jelle; Stewart, Robert

VERSION 1 - REVIEW

REVIEWER	Dorina Cadar University College London, UK
REVIEW RETURNED	05-May-2020

GENERAL COMMENTS	This is a case-control study examining comparatively body mass index, blood pressure, and serum total cholesterol levels between dementia cases and controls at multiple time intervals prior to dementia onset, using six European electronic health records databases. The manuscript is well written conveying clear messages and interpretation of the results. I only have a couple of minor recommendations. 1. I suggest a more detailed discussion of the time periods (0-2, 2-4, 4-6 years before diagnosis) when a more prominent change was observed for each of the outcomes of interest,2. Given that in England we read from left to write and the progression of the risk factors is monitored from 12 years before diagnosis closer to the time of diagnosis, I suggest a change in the order of the columns in Table S1 and Table S2 starting with 10-12 years and finishing with 0-2 years. That will mirror the construction of figures and the logical progression of time and outcomes as a patient approaches the time of diagnosis.
---

REVIEWER	Jaime Perales Puchalt University of Kansas Medical Center, USA
REVIEW RETURNED	06-May-2020

GENERAL COMMENTS	Thank you for inviting me to review the manuscript entitled "Vascular and metabolic risk factor differences prior to dementia diagnosis: a
--

	multi-database case control study using European electronic health records". This manuscript is remarkably well written, the dataset has high generalizability in the European context and the findings are interesting. I only have a few minor comments: Introduction: Please give a little more context about the types of data sources to convey the degree to which the sample represents the population (national or regional). For example, do they include only primary care from public service, or does it also include private service? Do all data except the Barcelona and Jutland include patients nationally? Is there a way to know what percentage of the population use primary care services included in those records? Methods: It would be nice to have a little summary if possible of how dementia was define, besides the reference. For example, does it include all dementias or only most common types (e.g. AD, VD)? Please justify the use of the term "gender" as opposed to "sex". Also, please use it consistently (the abstract says "sex") What is the rationale for including hypertension when you are assessing the association with DBP and SBP? What statistical package was used? What was the significance level for associations and interactions? Results: It would be nice to know the distribution of dementia by dementia type, but this has already been discussed in the limitations section. Might be interesting to mention missingness by region, for example, in Barcelona, S/DBP is not included and almost all BMI data is missing consistently across the years. It will be worth to point this regional pattern in missingness in the discussion section when talking about generalizability. Discussion: Might be worth discussing that different types might be associated differently with vascular factors. Information that we are missing due to the operationalization of dementia. Is there similar data from low- and middle-income countries that could support the conclusions or can we conclude the findings are generalizable to all populations? How do you think analyses have been limited due to the lack of control for variables such as educational level? "While people will have received a dementia diagnosis early in the course of the condition, others may receive this after several years of significant symptoms"- Is there a European estimate of the delay in dementia diagnosis? It would be interesting information to support this discussion. Do findings have any implication for dementia treatment/prevention?
--	---

VERSION 1 – AUTHOR RESPONSE

Reviewer: Please justify the use of the term "gender" as opposed to "sex". Also, please use it consistently (the abstract says "sex")	We have now changed the term gender to sex throughout the document	Page 5 onwards
---	--	----------------

Reviewer: Please give a little more context about the types of data sources to convey the degree to which the sample represents the population (national or regional). For example, do they include only primary care from public service, or does it also include private service?	We have added following section: In this study we used data from several European national or regional databases – both primary care general practitioner records and hospital based databases – containing a total number of over 22.5 million patients who used mainly public/ national health services.	Page 9 and further described in page 10
---	--	---

Reviewer: Do all data except the Barcelona and Jutland include patients nationally? Is there a way to know what percentage of the population use primary care services included in those records?	We now added following section: The THIN database contains data from participating UK primary care services with records from 1990 on around 9 million patients in total and 4 million receiving active care at a given time. Demographic profiles have been found to be similar to national estimates although THIN contains fewer people aged under 25 years; condition prevalence has also been found to be comparable[15]. The IPCI database is drawn from primary care records in Netherlands since 1990 on around 2.6 million patients in total, and 1.4 million receiving active care; its data are drawn from what has been found to be a representative sample (n=750) of general practitioners (GPs) in the Netherlands [16]. AUH is a regional record linkage system in Aarhus, Denmark containing both inpatient and outpatient records since 1989 with 1.8 million active patients, drawing data from different public bodies and including all ages [17]. IMIM-UPF contains hospital data from the Barcelona region with 1.6 million active patients and including all hospital admissions. HSD contains primary care records in Italy with nearly 2.3 million patient records from more than 800 participating primary care services throughout Italy: a longitudinal observational database that has been found to be representative of the general Italian population, established in 1998 by the Italian College of General Practitioners [18]. SIDIAP contains primary care records from Catalonia in Spain with nearly 6 million patient records from family paediatricians and general practitioner medical records and data from all ages [17].	Page 10
---	--	---------

Reviewer: It would be nice to have a little summary if possible of how dementia was define, besides the reference. For example, does it include all dementias or only most common types (e.g. AD, VD)?	We have included following statement: Cases were defined on the basis of a diagnosis of dementia and included all assigned dementia diagnoses of any type (inclusion criterion) and the index date was set as that on which the first dementia diagnosis was recorded.	Page 12
Reviewer: What is the rationale for including hypertension when you are assessing the association with DBP and SBP?	We have included following statement: Hypertension diagnosis was considered as a relevant covariate as patients with sustained hypertension in midlife have an increased risk for dementia regardless of their BP during late life [20] and because the interest was on changes in the BP relationship over time, which was not considered to be cotermious with a hypertension diagnosis.	Page 13
Reviewer: What was the significance level for associations and interactions?	We have included following statement: The significance level for all tests was defined as a p-value \leq 0.05.	Page 14
Reviewer: What statistical package was used?	We have included following statement: Analyses were carried out using IBM SPSS Statistics version 20.0.	Page 14

Reviewer: It would be nice to know the distribution of dementia by dementia type, but this has already been discussed in the limitations section.	This is limitation of the study and have discussed this in the discussion limitations section	Page 15
---	---	---------

Reviewer: Might be interesting to mention missingness by region, for example, in Barcelona, S/DBP is not included and almost all BMI data is missing consistently across the years. It will be worth to point this regional pattern in missingness in the discussion section when talking about generalizability.	We have included following statement: Supplementary table 2, summarises proportions of patients with missing data for each vascular risk factor in each database. In summary, AUH had limited BMI data, with no values prior to 6 years before the index date and no data for SBP and DBP. For HSD, 94% of BMI data were missing in the 10-12 year window before the index date, improving to 78% by the 0-2 year window and similar trends were observed for BMI data in the IPCI database, with 93% missing at 10-12 years and 67% missing data at 0-2 years before the index date. In the SIDIAP database there were no BMI, blood pressure or total cholesterol data at 10-12 years before the index date. Generally, compared with other databases, there were fewer missing data in the THIN database for all risk factors. IMM-UPF did not have blood pressure data, and had no BMI data at 10-12 years before the index date.	Page 15 amnd 16
---	--	-----------------

Reviewer: I suggest a more detailed discussion of the time periods (0-2, 2-4, 4-6 years before diagnosis) when a more prominent change was observed for each of the outcomes of interest,	We have included following statement: For BMI, the largest decline in meta-analysed case-control differences was observed between 2-4 years and 0-2 years before the index diagnosis of dementia. For SBP, the meta-analysed case-control difference increased from 8-10 years to 4-6 years and then declined steeply from 2-4 years to 0-2 years. For DBP, the largest meta-analysed case-control differences were observed in the 2-4 year and 0-2 year periods before the diagnosis of dementia. For total serum cholesterol levels, the meta-analysed case-control difference increased from 10-12 years to 6-8 years before the diagnosis of dementia and then declined up till the diagnosis of dementia with a slightly large decline just before diagnosis of dementia.	Page 16
---	--	---------

Reviewer: Might be worth discussing that different dementia types might be associated differently with vascular factors. Information that we are missing due to the operationalization of dementia.	We have included following statement: In previous research, there has been some suggestion of an age-dependent effect of blood pressure on Alzheimer disease, although the evidence is relatively weak: there was a suggestion that midlife diastolic, but not midlife systolic, hypertension in midlife may increase risk of incident Alzheimer disease and a suggestion that elevated late-life BP may actually be beneficial (Power et al., 2011). However, the same systematic review [37](Power et al., 2011) and meta-analysis of prospective epidemiologic research, did not provide clear evidence for a relationship between blood pressure and incidence of Alzheimer disease. On the other hand, [38] found higher SBP among patients with vascular dementia (OR per 10 mmHg: 1.33). Several well-established longitudinal studies have reported on the relationships between the trajectory of change over time in levels of blood pressure, cholesterol and BMI and incident all-cause dementia, with some additionally reporting on Alzheimer's disease, vascular dementia and MCI [39]. Authors found that the results for Alzheimer's disease, vascular dementia and MCI were similar but with fewer data points, with those who went on to develop all-cause dementia showing a greater increase followed by a sharper decrease in blood pressure and/or BMI before a positive diagnosis.	Page 21
Reviewer: Is there similar data from low- and middle-income countries that could support the conclusions or can we conclude the findings are generalizable to all populations?	We have included following statement: In a collaborative study of over 1.3 million adults from Europe, the United States, and Asia, higher BMI was associated with increased dementia risk when weight was measured >20 years before dementia diagnosis [40], but this association was reversed when BMI was assessed <10 years before dementia diagnosis and this finding is consistent with our study as well as a systematic review [41]. Furthermore, comparisons of risk factors for dementia, AD, and VaD, in developed and developing world regions were carried out in a review by Kalaria in 2010 which concluded that there was a high risk effect of factors such as vascular diseases and smoking for incidence of dementia in developed regions (North America, Europe, Japan) as well as in Asia (China, Guam, India, South Korea, Taiwan). Dementia was significantly associated with reported stroke and diabetes in logistic regression models adjusted for sociodemographic status and other vascular risk factors (OR (95%CI) 4.40 (2.70 to 7.19) and 1.56 (1.20 to 2.03), respectively) in a national survey of older people in Trinidad.	Page 22
Reviewer: How do you think analyses have been limited due to the lack of control for variables such as educational level?	We have included following statement: Information on education level was not available in these routine healthcare data resources. On the one hand, low educational achievement has been shown to be a robust risk factor for dementia[43]; on the other hand, intellectually stimulating, socially engaging, or physical activities might lower the risk of dementia [44]. The situation is not different in developing countries, where surveys have consistently identified low education as a risk factor for dementia [45]. However, in some communities, low literacy is often linked to poverty or lower socioeconomic status, which is also associated with poorer health, lower access to health care, and increased risk of dementia [45]. While it may well be the case that lower education confers both a higher risk of dementia and a higher risk of adverse vascular risk factors, it is less clear whether there should be any association between education and the changes over time in relationships between vascular risk factors and dementia, described in this study.	Page 22
Reviewer: Might be interesting to mention missingness by region, for example, in Barcelona, S/DBP is not included and almost all BMI data is missing consistently across the years. It will be worth to point this regional pattern in missingness in the discussion section when talking about generalizability.	We have included following statement: Furthermore, two databases, AUH and IMIM-UPF, did not contribute blood pressure data and IMIM-UPF and HSD had high proportion of missing data for BMI throughout the study period.	Page 23
Reviewer: "While people will have received a dementia diagnosis early in the course of the condition, others may receive this after several years of significant symptoms"- Is there a European estimate of the delay in dementia diagnosis? It would be interesting information to support this discussion.	We have included following statement: For example, the length of time between dementia symptoms being noticed and diagnosis of dementia has been estimated to vary in Europe from 1.61 years in Italy to 2.49 years in Scotland and 2.57 years in Netherlands [46].	Page 23

Reviewer: Do findings have any implication for dementia treatment/prevention?	We have included following statement: In examining the patterns and trajectories of the established risk factors for cognitive decline and dementia, this study adopted a relatively broad approach to an emerging area of enquiry. Our study focused on three risk factors of interest that were envisaged to be most available (blood pressure, cholesterol and BMI); however, future work could feasibly include greater numbers of risk factors and the interaction between different individual changes in status (for example the combination of changes in blood pressure and BMI), which could eventually lead to more personalised risk assessments and targeted interventions early in the asymptomatic, prodromal phase of cognitive decline and dementia.	Page 24
Reviewer: Given that in England we read from left to write and the progression of the risk factors is monitored from 12 years before diagnosis closer to the time of diagnosis, I suggest a change in the order of the columns in Table S1 and Table S2 starting with 10-12 years and finishing with 0-2 years. That will mirror the construction of figures and the logical progression of time and outcomes as a patient approaches the time of diagnosis.	We have now made changes to Supplementary table 1 and 2 as requested.	Supplementary table 1 and 2

VERSION 2 – REVIEW

REVIEWER	Jaime Perales Puchalt University of Kansas Medical Center, USA
REVIEW RETURNED	17-Jun-2020

GENERAL COMMENTS	All my comments have been successfully addressed. One very minor comment: "The was no specific patient or public involvement in the design or interpretation of this particular study."- "The" should be "There".
--